# A System Dynamics Model of Online Stores' Sales: Positive and Negative E-WOM and Promotion Perspective

**Qiang Yan, Simin Zhou \***  , **Xiaoyan Zhang and Ye Li**

School of Economics and Management, Beijing University of Posts and Telecommunications,
Beijing 100876, China; yan@bupt.edu.cn (Q.Y.); 2013213376@bupt.edu.cn (X.Z.); ye.li@se11.qmul.ac.uk (Y.L.)
\* Correspondence: zhousimin@bupt.edu.cn

**Abstract:** In this paper, we build a causal interaction diagram between the factors that may influence the sales and profits of online stores. An online store's real operation data were used to help determine the causal relationship between variables. Finally, we proposed a system dynamics model and conducted a simulation of the operation of an online store. In this model, we focused on the impact of promotion and positive/negative electronic word of mouth (e-WOM) on the sales and profits of the online stores. The simulation results showed a similar trend to the real data and the main research finding showed that promotion is not a long-term measure for the sustainable development of online stores. Excessive promotion effort may lead to consumers' dissatisfaction leading the increase of negative e-WOM. The systematic simulation can help us understand better the long-term effect of promotion and e-WOM on the operation of online stores. Finally, we gave some management suggestions for online stores' sustainable operations.

**Keywords:** online store; positive e-WOM; negative e-WOM; promotion; system dynamics; sustainable operation

## 1. Introduction

With the development of the internet, the emergence of electronic commerce has broken the limitations of traditional retail. Today, online transactions occupy a large part of our life. In a fiercely competitive environment, it is quite critical for online stores to improve the sales and profits for sustainable development. In virtual network transactions, we have no access to see the actual goods, so the trust and evaluations of online consumers becomes quite important. Consumers can create much more value for the company, not only economic value, but also intangible value including electronic word of mouth (e-WOM), referees, and feedback [1], which give an essential reference for other consumers.

In recent years, researchers have paid a lot of attention to the impact of e-WOM on online sales. Consumers can collect information about products or services and share their own opinions, evaluations, or knowledge on the internet [2]. Research shows that e-WOM has a significant impact on consumers' brand cognition, preference, perceived value, purchase intention, and decision making [3,4]. Therefore, e-WOM is extraordinarily vital for the operation and development of an enterprise. Positive e-WOM can promote sales of the enterprise [5], while the spread of negative e-WOM can quickly lead to a crisis [6].

In addition to the e-WOM towards a product, the product price is also an important factor that contributes to online stores' sales and profits, while it is the promotion measures taken by the online stores that influence the price fluctuations. Promotions have a specific impact on the perceptions,

attitude, and behaviors of the consumers [7]. Research has shown that promotions have a short-term increase effect on the sales of products [8]. Sometimes, promotions can also cause some adverse impacts, such as price fatigue and lower perceived quality. In retrospect, the server congestion and lower quality of service during the Double eleven sales activities of a Tmall shop in China caused great consumer dissatisfaction.

Nowadays, there is a lot of literature about the impact of positive/negative e-WOM and promotions on consumer decision making and online stores' operation. However, there is a lack of studies that focus on the interactions of the factors that are relevant to the operation of online stores. There needs to be systematic discussion to discover the relationships between these factors. Meanwhile, the study methods of previous studies were limited. Existing studies mainly used questionnaires to obtain research data, which lacked representativeness compared to real data.

Therefore, in order to fit the research gaps, this paper uses the system dynamic method to structure a model of online stores' operations and especially consideres the impacts of e-WOM and promotion. The real operation data of an online store was used to help determine the causal relationships between variables and was compared with the simulation results. The following are contributions of this paper: (a) we build a framework for the operation of online stores, which provides a more comprehensive perspective to understand e-commerce, (b) we consider the systematic impact of positive and negative e-WOM on online stores' operation, which were ignored in previous studies, and the findings update the context of e-WOM to some extent, and (c) the real data used in this study can complement the empirical research methods used in previous research. The research findings provide some reference for the sustainable operation and management of online stores and help them predict sales volume in the future.

## 2. Literature Review

### 2.1. Electronic Word of Mouth

Electronic word of mouth (e-WOM) refers to the content generated by users, which can effectively compensate consumers for perceived risk, asymmetric information, and lack of trust. More and more consumers will search related details before buying. Consumers can obtain vast quantities of information from e-WOM. This information provides an essential reference for the consumer's purchase decision. Therefore, e-WOM has an enormous impact on the sales of enterprise products, and it has attracted wide attention from enterprises.

Previous studies have confirmed the impact of e-WOM on consumers [9–11]. Compared with the traditional word of mouth, e-WOM has stronger diffusivity, longer duration, and is easier to obtain and measure. It also has higher persuasiveness and credibility [12–14]. Researchers show that e-WOM can affect consumers' perception of brand, preference, and purchase decision [3,4]. The volume, quality, and validity of online reviews have a positive impact on the consumer's willingness to purchase [15], and the impact is persistent. Kim et al. found that when the number of positive or negative e-WOM grows [16], the effect of e-WOM on profits would change. Bickart & Schindler have found that a positive e-WOM can promote the sales in a short-term [5], while negative e-WOM may reduce sales, and the higher the intensity, the higher the influence [6]. E-WOM has an immeasurable impact on the market.

There is a significant positive effect between the e-WOM and enterprise performance [17]. On the one hand, e-WOM will help the enterprises understand the consumer response to the product and accordingly improve the quality of the product. On the other hand, positive e-WOM has a positive impact on persuading consumers to trust brands and buy products. Negative e-WOM will cause damage to the image of products and enterprises. Therefore, it is necessary to study the impact of the double sided e-WOM and help enterprises with enhancing the consumer trust and purchase intention.

*2.2. Sales Promotion*

In the online purchase scenario, consumer evaluations, price, and promotion activities have become important references for consumers when they select from the various goods. Promotion is a useful marketing tool, through which an enterprise can commit consumers to its brands more efficiently [18].

Studies have pointed out that the sales promotion can significantly enhance people's perception of value [19], encourage consumers to buy goods or services, and stimulate sales in the short term [20]. However, oversized promotional efforts will increase the cost and bring a specific burden to the development of the enterprise [21]. Besides, once consumers get used to the promotional discount, they will have a negative attitude towards the original price, and then buy other products [22]. Mela et al. suggested that frequent price promotions could increase customers' price sensitivity [23], ultimately resulting in reduced brand loyalty and lead customers to be negative about the price. As a result, sales promotion is a short-term way to increase sales volume for online stores, and from a long-term perspective, it may bring a decline in profits and lower consumer evaluations. Therefore, it is particularly important for online stores to adopt a suitable promotion method. In this paper, we mainly consider the price promotion.

## 3. System Dynamic Model

*3.1. System Boundary*

As the online store's sales system involves a lot of variables, it is difficult to consider all of them. The boundary of the system studied depends on the research problem of the study. In this paper, we observed the case of an online store's sale from the perspective of operations, so we will not discuss the influence of consumers' characteristics on sales. We focused on the systematic influences of the e-WOM and promotion measures on the sales volume and profits of the online store. Those variables that have no significant impact on the observed variables were put outside of the model. According to the research, there are some variables related to the promotions measures: price, service quality, and the variables related to the e-WOM—positive and negative reviews volume, product quality, service quality, post-purchase evaluation, e-WOM maintenance, and maintenance cost. In the two-month simulation period, some fixed costs such as employee wages had almost no change, so they were not considered in the system model. Therefore, the online store's sales system mainly involves the variables listed in Table 1.

*3.2. Causal Interaction Diagram*

In general, sales volume and profits volume can reflect the operations of an enterprise. In order to take appropriate marketing measures and maintain sustainable development, online stores should learn how to predict the future trend of sales and profits. Research shows that decrease of product price has a significant and immediate effect on promoting the sales and increasing market share. As a result, an increase in promotion efforts makes sales volume increase to some extent (positive feedback). Accordingly, the rise in price will reduce the sales volume (negative feedback), while increasing promotion efforts will reduce the quality of business service (negative feedback) and the increase in promotion cost (positive feedback).

At the same time, the improvement of consumer evaluation (e-WOM) towards the products will increase sales volume by increasing other consumers' willingness to purchase. Therefore, the volume of positive reviews on the e-commerce website will increase online stores' sales volume (positive feedback), while the increase in negative reviews will reduce sales volume (negative feedback) [24]. Conversely, the increase in sales will increase the number of positive and negative reviews on the website (positive feedback). Besides, consumers' psychological factors will also influence their e-WOM sharing. This paper uses consumers' post-purchase evaluation to reflect the psychology of the consumers. When the comprehensive evaluation of consumers increases, the number of positive

reviews on the website will increase (positive feedback), while the number of negative reviews will decrease (negative feedback) [25]. There are many factors that may influence consumers' evaluation. The higher the product quality and service quality, the higher the consumers' evaluation will be (positive feedback) [26], while a higher price will reduce the post-purchase evaluation (negative feedback).

**Table 1.** The explanation of the system modeling variables.

| Nomenclature | |
| --- | --- |
| **Level variable** | |
| Daily sales | The volume of daily sales in the current simulation circle (CSC). |
| Profits | The accumulated profits of the online store in the CSC. |
| Post-purchase evaluation | The average value of consumers' post-purchase evaluations in the CSC. |
| **Rate variable** | |
| Sales increase | The increase in volume of sales compared to the last simulation cycle (LSC). |
| Sales decrease | The decrease in volume of sales compared to the LSC. |
| Income | The daily total income of the online store in the CSC. |
| Cost | The daily total cost of the online store in the CSC. |
| Evaluation increase | The average evaluation increase of the consumers compared to the LSC. |
| Evaluation decrease | The average evaluation decrease of the consumers compared to the LSC. |
| **Auxiliary variable** | |
| Daily positive reviews | The amount of daily positive reviews of consumers in the CSC. |
| Daily negative reviews | The amount of daily negative reviews of consumers in the CSC. |
| Price | The price of the product in the CSC. |
| Promotion effort | The promotion effort taken in the CSC. |
| Promotion cost | The cost of promotion in the CSC. |
| E-WOM maintenance | The effort of e-WOM maintenance given by the online store in the CSC. |
| Maintenance cost | The cost of e-WOM maintenance in the CSC. |
| Product quality | The quality of product sold by the online store in the CSC. |
| Service quality | The quality of service provided by the online store in the CSC. |

Since negative reviews will affect sales, online stores need to reduce adverse impact by e-WOM maintenance (positive feedback). E-WOM maintenance will not only improve consumers' post-purchase evaluation (positive feedback) but also increase maintenance cost (positive feedback), which result in a lower profit (negative feedback).

Based on the above discussion, we draw the causal interaction diagram of online stores sales system in Figure 1.

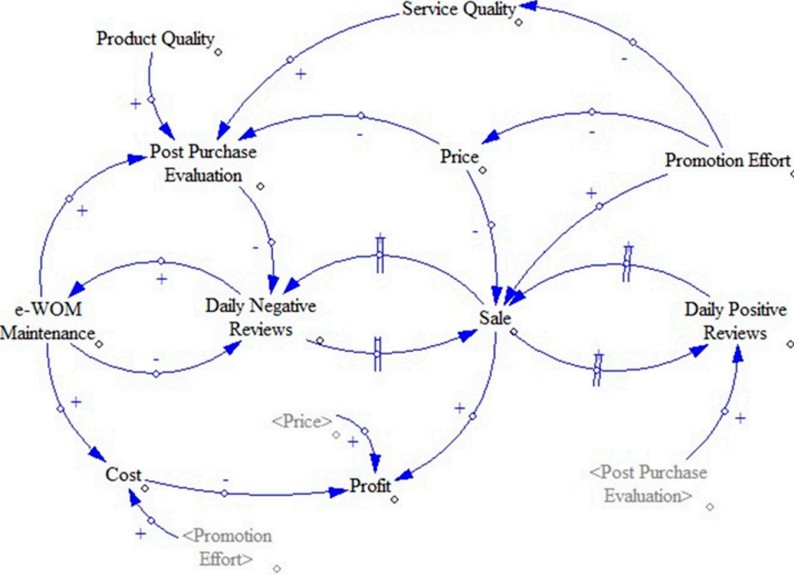

**Figure 1.** Causal interaction diagram of the online store sales system. Notes: the "+" represents positive feedback; the "−" represents negative feedback.

*3.3. Feedback Loop Discussion*

In the causal interaction diagram above, there are several main feedback loops.

(1) Promotion feedback loop: this subsystem is a "drinking poison to quench thirst" model. On the one hand, promotion can increase profits through the increase of sales. On the other hand, it will reduce the service quality and the evaluation of consumers, and accordingly reduce sales and increase the cost.

(2) E-WOM feedback loop: the feedback loop between reviews and sales is a growth upper limit model. The increase in sales volume can increase the number of positive reviews and then return to increase sales volume. However, the increase in sales volume may also increase the number of negative reviews, thus reducing sales. Therefore, enterprises need to reduce the number of negative reviews through external measures, such as e-WOM maintenance.

(3) Marketing feedback loop: e-WOM maintenance is a marketing tool for online stores. This subsystem is a "drinking poison to quench thirst" model. This means that e-WOM maintenance can enhance corporate reputation and increase sales volume on the one hand, and it will increase operation cost and bring down profits on the other hand.

Therefore, sales promotion and e-WOM maintenance are short-term strategies. The most fundamental means for an enterprise to increase sales and profits are improving product quality and service quality.

*3.4. Stock and Flow Diagram*

According to the causal interaction diagram of the online stores sales system, we can draw a corresponding system stock and flow diagram (Figure 2).

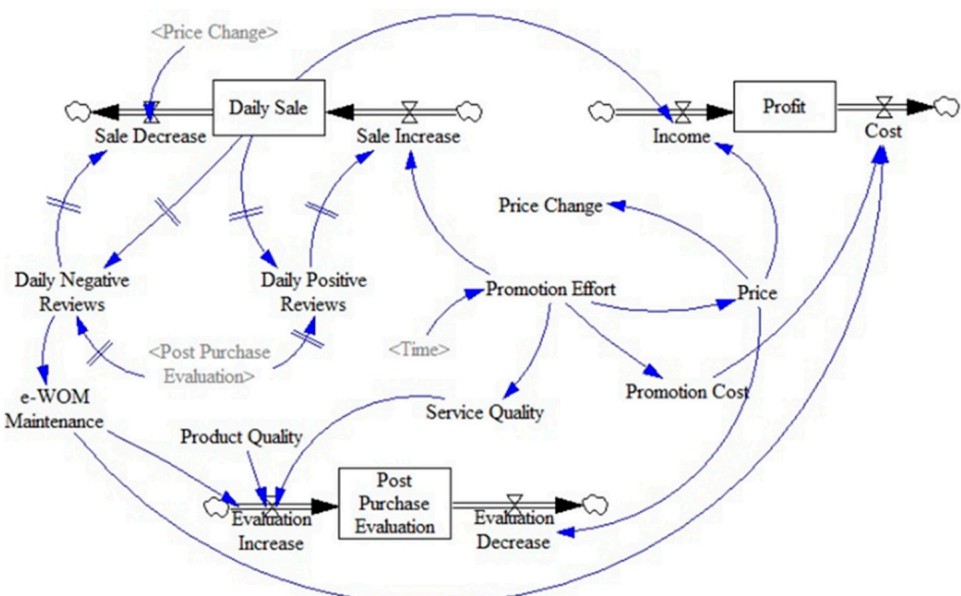

**Figure 2.** Stock and flow diagram of the online store sales system.

In the above stock and flow diagram, the post-purchase evaluation, profit, and the daily sales are stock variables. The income, cost, sales increase, sales decrease, evaluation increase, and evaluation decrease are flow rate variables, and the other variables are auxiliary variables.

Since our simulation's time span is two months, some variables such as employee wages are a constant in this period. We excluded this constant variable from our model to make it simple and effective. Also, consumer reviews are usually generated after purchase. Therefore, we set some delay equations in the system model.

## 4. Result Analysis

### 4.1. Data Collection

We collected the reviews and sales data of a product from a Tmall supermarket daily. The data collection process was supported by crawler program with Python 3.5. The time span was from 5 October 2017 to 5 December 2017. The product was type of daily necessity with thousands daily turnovers. The system dynamic simulation was performed on Vensim. The time internal was set to 1 day to simulate the sales and profits trend. The daily sales and the e-WOM maintenance are were all the real data obtained from the website and the numbers of positive and negative e-WOM were based on the score calculated by Python sentiment analysis. If the score was greater than 0, the piece of e-WOM was regarded as positive e-WOM, otherwise, the e-WOM was negative. In the simulation model, we inputted the data about daily promotion effort based on the real data and then observed the fluctuations of daily sales and accumulated profits. To determine the correlation coefficient between variables in the model, we firstly normalized the data, and then used SPSS 23 to do a regression analysis. Finally, we compared the simulation results with the real data to verify the effectiveness of the model.

### 4.2. Simulation Equations

We used equations to describe the relationships between system variables. The relationships between variables and the correlation coefficients were determined by results of the regression analysis. The calculation equations of the main variables follow, and the value of index is show in Appendix A.

Daily sales (*DS*) represents the daily sales volume of the product sold by the online store. The *DS* is defined by a time integral of the sales increase (*SI*) and sales decrease (*SD*). The initial value of *DS* was set by the daily sales volume on 5 October 2017.

$$DS(t) = \int_{t0}^{t} (SI(t) + SD(t)) + DS(t_0) \tag{1}$$

Then we defined the equation of daily *SI* and *SD*. In the literature review section, we conclude that promotion effort and e-WOM will influence online stores' sales volume. Therefore, sales increase (*SI*) volume was defined by daily promotion effort (*DPE*) and daily positive reviews (*DPR*). Furthermore, the *DELAY*1 function was used due to the time (*RET*) required to post reviews.

$$SI(t) = \alpha_1 * DPE(t) + DELAY1(\alpha_2 * DPR(t), RET) \tag{2}$$

The equation of sales decrease (*SD*) is similar to the *SI*. The increase in price (*PC*) and daily negative reviews (*DNR*) will directly influence the sales volume. This is defined in the following equation.

$$SD(t) = \beta_1 * PC(t) + DELAY1(\beta_2 * DNR(t), RET) \tag{3}$$

The value of daily promotion effort (*DPE*) is determined by the online store, and it varied with the time. Therefore, *WITHTIME* function was used in this equation. In this simulation model, there are five levels of *DPE* (1–5). Level 1 represents the original price of the product (*Pr*) and level 5 means the price is a 50% discount. The value of price change (*PrC*) was calculated by the product price in the current simulation circle minus the product price in last simulation circle.

$$DPE(t) = WITH(TIME, [(0,0) - (60,5)]) \tag{4}$$

$$Pr(t) = WITH(DPE, [(0,0) - (5,60)]) \tag{5}$$

$$PrC(t) = Pr(t) - DELAY1(Pr(t), 1) \tag{6}$$

The number of daily positive (*DPR*) and negative reviews (*DNR*) was influenced by the post evaluation of consumers (*PPE*) and the daily sales (*DS*) volume.

$$DPR(t) = DELAY1(\theta_1 * PPE(t), RET) + DELAY1(\theta_2 * DS(t), RET) \tag{7}$$

$$DNR(t) = DELAY1(\varepsilon_1 * PPE(t), RET) + DELAY1(\varepsilon_2 * DS(t), RET) \tag{8}$$

As for the post-purchase evaluation (*PPE*) of consumers, on the one hand, evaluation increase (*EI*) will be influenced by the product and service quality (*PQ* and *SQ*), and the e-WOM maintenance measure (*EM*) taken by online stores. On the other hand, the increase in price (*Pr*) will result in evaluation decrease (*ED*).

$$PPE(t) = \int_{t0}^{t} (EI(t) + ED(t)) + PPE(t_0) \tag{9}$$

$$EI(t) = \eta_1 * EM(t) + \eta_2 * PQ(t) + \eta_3 * SQ(t) \tag{10}$$

$$ED(t) = \delta_1 * Pr(t) \tag{11}$$

The online store will take e-WOM maintenance (*EM*) measures according to the negative reviews posted by the consumers, so the value of *EM* was determined by the *DNR*. In the equation $EM_0$ represents the fixed maintain value.

$$EM(t) = \gamma_1 * DNR(t) + EM_0 \tag{12}$$

As for product quality and service quality, the *PQ* was constant in our study since the simulate cycle was only two months, while the service quality (*SQ*) was influenced by the promotion effort (*PE*).

$$SQ(t) = \gamma_2 * PE(t) + SQ_0 \tag{13}$$

The accumulated profit (*AP*) of the online store is the integral of income (*IC*) and cost (*CS*). The daily *IC* is equal to the price (*Pr*) multiplied by daily sales volume (*DS*) and the *CS* consists of the maintenance cost and promotion cost.

$$AP(t) = \int_{t0}^{t} (IC(t) + CS(t)) + AP(t_0) \tag{14}$$

$$IC(t) = Pr(t) * DS(t) \tag{15}$$

$$CS(t) = \omega_1 * EM(t) + \omega_2 * DPE(t) \tag{16}$$

*4.3. Simulation Results*

4.3.1. Sales Simulation

Through system simulation, the fluctuations of the product sales are shown in Figure 3. The blue line indicates the trend of real sales data, and red line shows the trend of simulation sales volume. We used relative error (Equation (17)) to represent the accuracy of the simulation result. As the relative error was 4.17%, we can conclude that the simulation result and the real data fit well. When maintained at a stable price and promotion level, the sales volume changes were small. At day 8 and 23, the daily sale showed a downward trend, which was due to the price-raising measures taken in those two days. Beginning from point 38, daily sales rose sharply, until point 42, when they started to decline. It was observed that during this period, the online store participated in the Double eleven promotional activity, and the promotion effort was much larger than before, which led to a rapid increase in sales volume. At point 44, the online store recovered the original price, but the sales volume did not decrease to the same level as before, which indicated that there were some factors that may influence sales more than just promotion and price. Sales increase and price decrease had caused an increase in

positive reviews on products, in turn, it promoted consumer purchase and resulted in higher sales level than before.

$$Relative\ Error = \frac{\sum |Simulation\ Value - Actual\ Value|}{\sum Actual\ Value} \times 100\% \tag{17}$$

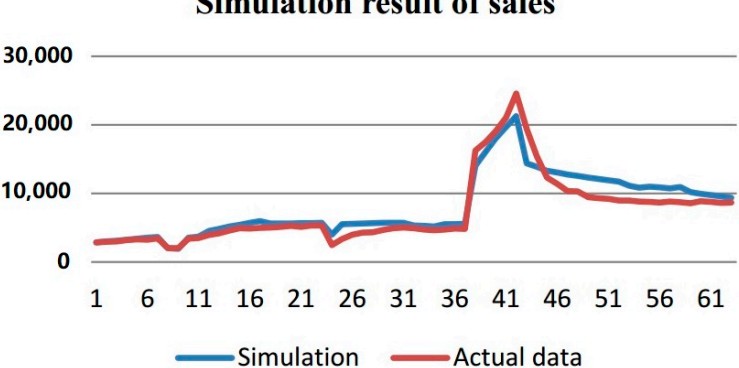

**Figure 3.** Sales Simulation result compared to the actual data.

Figure 4 shows a parameter control analysis, where curve 1 shows the variation trend of sales when promotion level = 1, curve 2 shows the trend when promotion level = 2, and curve 4 shows the trend when promotion level = 4. We assume that the online store maintains the same promotion effort level from points 1 to 60, while from point 60 to 300, the online store takes different levels of promotion effort (1 to 4) respectively. We can see that the sales volume of products maintains a steady upward trend after point 60, and the larger the sales promotion effort, the faster the sales volume grows. Therefore, sales promotion effort will indeed help in improving sales in the short term.

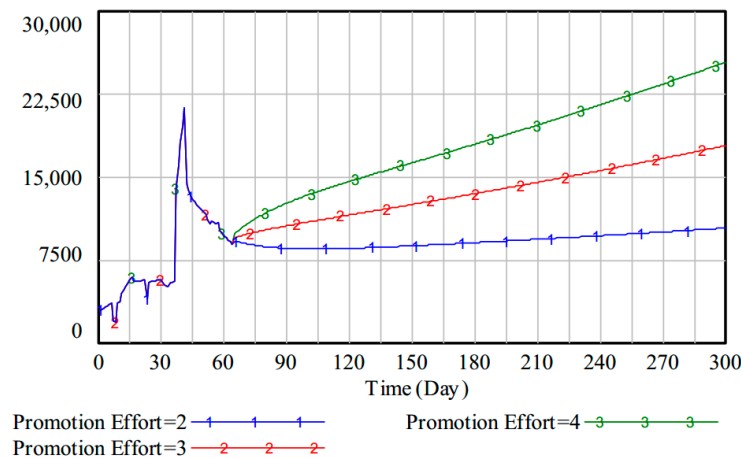

**Figure 4.** Different effect of promotion efforts' on sales.

4.3.2. Profit Simulation

Figure 5a shows the result of the online store's profit simulation. The profit level grew steadily but in the period from days 38 to 42, the sales promotion reached the maximum level, and profit increased slowly, and even had a decreasing trend. This indicates that the higher promotion level will not only improve the sales volume but also increase the cost. Therefore, in the operation process of online stores, they must constantly adjust the promotion strategies according to the situation in order to maximize profits. For example, if the profits of the online store start to decrease during the promotion period,

the promotion should be stopped in time and the improvement made should refer to the e-WOM of consumers.

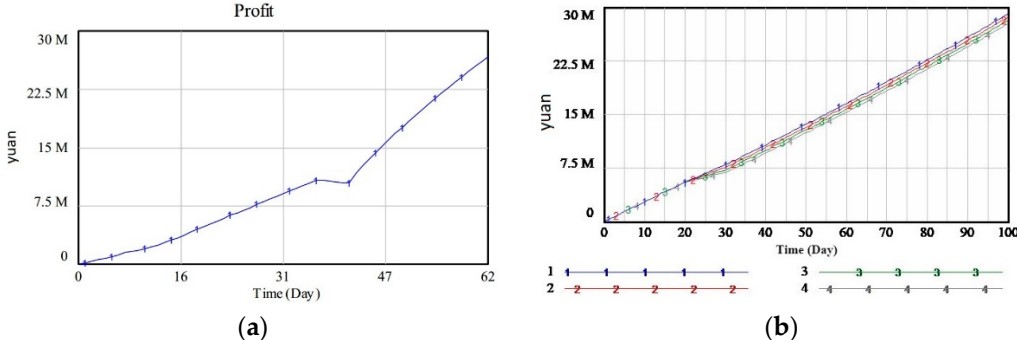

(a)        (b)

**Figure 5.** Simulation result of profits. (**a**) Profits result in real promotion effort data and (**b**) influence of different promotion efforts on online store profits.

Figure 5b shows a parameter control analysis of online store's profits when the promotion effort is different (level 1 to 4). The online store's profits show a continuous rising state. However, the higher the promotion effort, the slower the profit rise. This may be due to the higher promotion and e-WOM maintenance costs. Exorbitant promotion effort may lead to a decrease of service quality, which will make consumer evaluation decrease. Correspondingly, the daily sales volume may decrease, which leads to profit decrease. In order to verify the research findings, we explored the change trend of consumer post-purchase evaluation, and the results are shown in Figure 6.

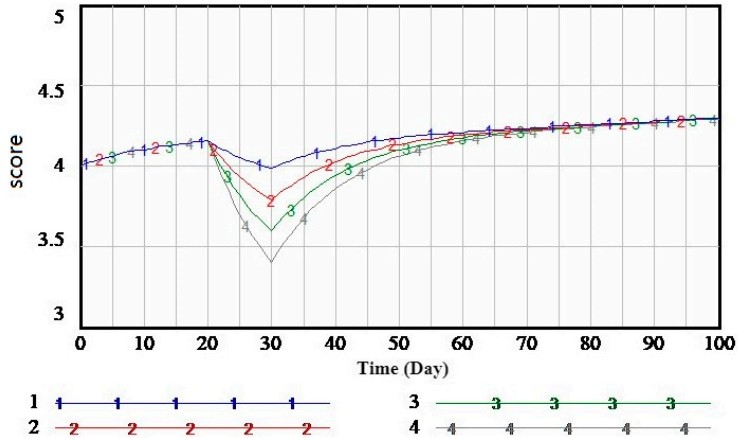

**Figure 6.** Influence of different promotion effort on consumers' evaluation.

Figure 6 shows the simulation trend of consumers' post-purchase evaluation under different promotion efforts. The results show that from day 20, the evaluation of consumers begins to decline after the online store start to take promotional activities. The greater the promotion effort, the faster the evaluation of consumers decrease. This may be due to the excessive promotion efforts leading to more consumers, so that the service quality has been reduced, resulting in consumer dissatisfaction. From point 30, the promotion activity has ended. Consumer evaluation began to rise, and finally stabilized. Therefore, the promotion was not entirely beneficial to the development of the online shopping mall, and there were still some negative effects.

In order to explore the factors that affect profit of the online store, we also analyzed the impact of different product quality levels on the online store's operation. The result is shown in Figure 7. We set the parameter values of product quality to 4.5, 7.5, and 9, respectively, and observed the simulation results of the profit. It can be seen from the Figure 7 that the higher the quality of goods, the faster the profit will rise.

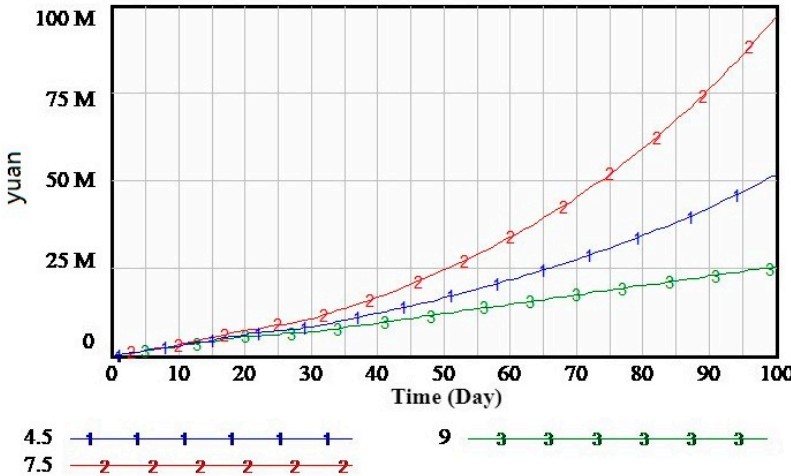

**Figure 7.** Influence of different promotion effort on consumers' evaluation.

### 4.3.3. E-WOM Simulation

Figure 8 is the simulation result of positive and negative e-WOM. Compared the Figures 3 and 8, we found that the variation trend of positive and negative reviews was similar to that of sales. The number of positive reviews was more than the number of negative reviews. Also, from the simulation results, the number of positive reviews was maintained at a level of 3 to 4 times than the number of negative reviews before Double eleven. From day 38 to day 42, the number of positive reviews was only about two times than the number of negative reviews. It can be seen that during the period of Double eleven, the number of consumers' negative reviews had increased. It may be that the low service quality caused a low evaluation. It also confirms that the high promotion level is not always suitable for the development of online stores.

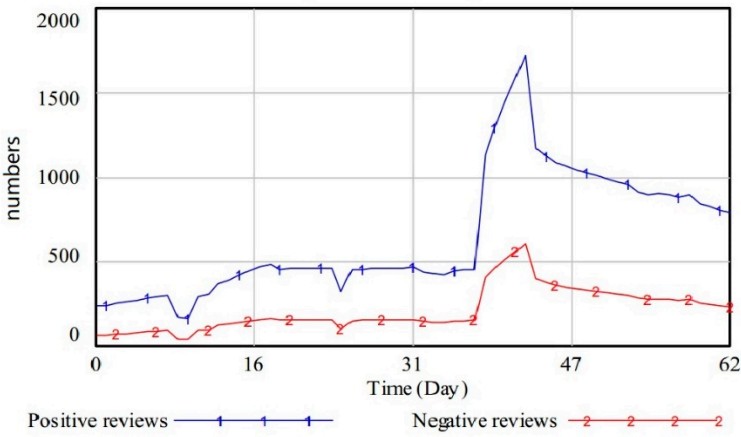

**Figure 8.** Simulation results of positive and negative reviews.

In addition, we divided the review data into two parts (ie., before and during Double eleven) to analyze the word frequency during the two period. We first used the third-party package (jieba) in Python to segment consumers' reviews, and then count word frequency. The wordcloud picture was drawn by the wordcloud package in Python (Figure 9). It turns out that most of the reviews before Double eleven are more positive. High frequency words included "pretty good", "ultrafast", "very good", "satisfied", "tangible benefits", and some other positive words. During the Double eleven period, words such as "Slow delivery", "Service quality", "Product quality", and "Speed" have become high-frequency words. This shows that in the period of Double eleven, consumers were more concerned about the service quality and the delivery speed. The results showed that the quality of

service in the period of Double eleven had indeed declined, which caused consumer dissatisfaction and lower evaluations.

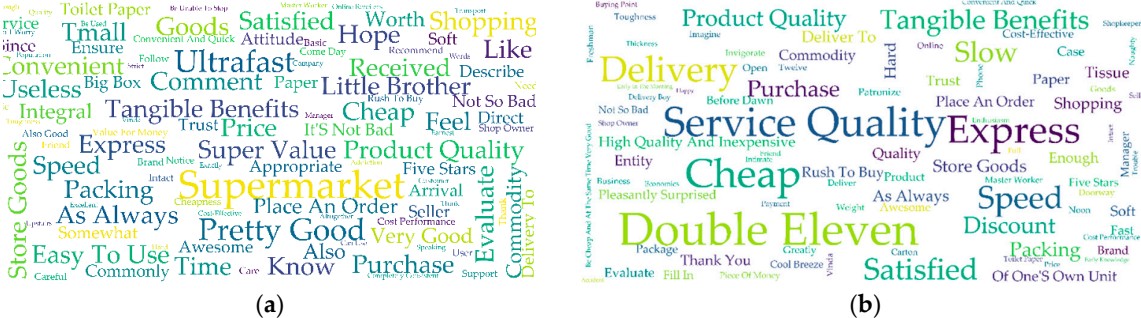

**Figure 9.** Word cloud pictures of reviews. (**a**) Word cloud before Double eleven and (**b**) word cloud after Double eleven.

## 5. Conclusions

With the development of the internet, e-WOM has become more and more critical for the operation and performance of enterprises, especially the online store, which are incredibly dependent on reputation and trust. For online stores, sales volume and profits are closely related to sustainable operation and development. According to the operation of online stores, we built a systematic model which included the variables that may influence the sales and profits of online stores. In this model, we focused on the impact of e-WOM and promotion on operation of online stores. This article provides an overall view of online stores' sustainable development. We used the real data of one online store's product to determine the relationship and coefficient between variables and then we conducted a simulation and compared the simulation results with the real data to validate the effectiveness of the system model. The final simulation results fitted the actual data well, which means that the simulation model is valid.

Firstly, from the simulation results, we found that the sales volume of online stores increases with increasing of the promotion level, but in the period of the Double eleven, the sales promotion reached the maximum level, and the profit increased slowly, or even had a decreasing trend. Secondly, we found that consumers' evaluations decreased with the increase of promotion effort. This may be due to the reduction of service quality caused by excessive promotion effort. Figure 8 shows that the number of consumers' negative reviews increased compared to the positive reviews. Hence the promotion effort may lead to negative e-WOM towards the online store whereas the improvement in product quality can indeed bring positive e-WOM. Finally, from the cloud picture in Figure 9, we found that during the period of Double eleven, consumers are more concerned about the service quality of the online store and the delivery speed. This finding further confirmed the previous findings that promotion effort may cause consumers' dissatisfaction.

Therefore, whether from the perspective of profit, consumer evaluation, or service quality, online stores cannot blindly promote sales by price promotion. From the long-term perspective, promotion is not suitable for the sustainable development of online stores. Online stores should constantly keep an eye on the consumer e-WOM, especially negative e-WOM. According to the feedback of consumers, online stores can discover their shortcomings and make corresponding improvements. Since the sales and profits are directly related to the consumer e-WOM, the product quality and service quality should be focused on.

In a word, the proposed system dynamics model can help enterprises understand the systematic impact of e-WOM and promotion on sales and profit. In addition, it can help companies predict future trends and accordingly make adjustments. System dynamics helps to establish a holistic perspective of online stores operation and give great implications for sustainable development.

## 6. Limitations and Further Research

There are still some shortcomings in this study. Firstly, the simulation unit of the system dynamics model in this paper is "day," and the simulation span is two months. In the simulation period, many variables do not change significantly, so the system model is not conducive to a long-term prediction. Therefore, in our future work, we will use comprehensive data to simulate, in order to improve and perfect the model. Secondly, the system dynamic model of this paper mainly considered the impact of consumer reviews and price promotions on the online stores' sales and profits. There are many other factors not taken into consideration, such as platform characteristics and other online stores' competition. In the future, we will consider more influence factors as much as possible to make the model more perfect and credible.

**Author Contributions:** Conceptualization, Q.Y. and S.Z.; methodology, S.Z. and Y.L.; data collection, S.Z.; software, S.Z. and Y.L.; validation, S.Z. and X.Z.; visualization, X.Z.; writing—original draft preparation, S.Z.; writing—review and editing, Y.L. and X.Z; project administration, Q.Y.; funding acquisition, Q.Y.

**Funding:** This research was funded by MOE (Ministry of Education in China) Project of Humanities and Social Sciences, grant number 16YJA630063; the CMCC (China Mobile Communications Corporation), grant number MCM20170505; the BUPT Excellent Ph.D. Students Foundation, grant number CX2018116.

**Conflicts of Interest:** The authors declare no conflict of interest.

## Appendix A

The initial values of the simulation and the values of the index are show in Table A1.

**Table A1.** Values of the index in the simulation.

| Parameter | Value | Parameter | Value |
|---|---|---|---|
| $\alpha_1$ | 109.34 | $\delta_1$ | 0.02 |
| $\alpha_2$ | 0.12 | $\gamma_1$ | 15 |
| $\beta_1$ | 1.62 | $\gamma_2$ | −0.2 |
| $\beta_2$ | 101.23 | $\omega_1$ | 1.14 |
| $\theta_1$ | 1 | $\omega_2$ | 200 |
| $\theta_2$ | 0.05 | $DS(t_0)$ | 2875 |
| $\varepsilon_1$ | 2 | $PPE(t_0)$ | 4 |
| $\varepsilon_2$ | 0.03 | $EM_0$ | 1000 |
| $\eta_1$ | 0.0002 | $SQ_0$ | 10 |
| $\eta_2$ | 0.1 | $AP(t_0)$ | 0 |
| $\eta_3$ | 0.02 | | |

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
