# Peer review of "A System Dynamics Model of Online Stores’ Sales: Positive and Negative E-WOM and Promotion Perspective"

_sustainability, doi:10.3390/su11216045_

Round 1
Reviewer 1 Report
This is an interesting paper on electronic word-of-mouth (eWoM) and promotion for an online store. To my knowledge, the paper cites all of the most relevant sources and builds a convincing conceptual model. The paper is unusual in drawing results from real, up-to-date data, which leads to a useful (though modest) contribution. The contribution could be clarified and emphasized more.
As far as I understand it, the simulation closely matches the real data. The simulation indicates that promotion has a short-term positive effect on sales but over time, there is no useful positive effect on profitability. This is but a modest contribution as that finding has been known in marketing literature for many decades. What is a new contribution here is the updated context of eWoM.
The title suggests an analysis of the effects of positive and negative eWoM but this is not delivered in the Conclusions section. The contributions in this regard and with respect to promotion should also be emphasised in the abstract.
The heading for Section 6, ‘Patents’, seems strange. I would have expected the heading ‘Limitations and further research’.
Author Response
Dear reviewer,
We are grateful to your professional suggestions. Based on these suggestions, we have made careful modifications on the manuscript. The modifications in our manuscript were all highlighted by using the revision mode of MS word.
Point 1: This is an interesting paper on electronic word-of-mouth (eWoM) and promotion for an online store. To my knowledge, the paper cites all of the most relevant sources and builds a convincing conceptual model. The paper is unusual in drawing results from real, up-to-date data, which leads to a useful (though modest) contribution. The contribution could be clarified and emphasized more.
As far as I understand it, the simulation closely matches the real data. The simulation indicates that promotion has a short-term positive effect on sales but over time, there is no useful positive effect on profitability. This is but a modest contribution as that finding has been known in marketing literature for many decades. What is a new contribution here is the updated context of eWoM.
Response 1: Thanks for your supportive comment and useful suggestion. We have modified the description of contributions in the Section 1. For your convenience, we list it below.
The contributions of this paper are following: (a) we build a framework for the operation of online stores, which provides a more comprehensive perspective to understand e-commerce; (b) we considered the system impact of positive and negative e-WOM on online stores’ operation which were ignored in previous studies, and the findings update the context of e-WOM to some extent; (c) the real data used in this study can complement the empirical research methods used in previous researches.
Point 2: The title suggests an analysis of the effects of positive and negative eWoM but this is not delivered in the Conclusions section. The contributions in this regard and with respect to promotion should also be emphasised in the abstract.
Response 2: Thanks for your suggestion. We have reorganized the Conclusion section and added more descriptions of research findings in our paper. Besides, we emphasized our contributions in the abstract. Also, we list the improved content below.
Conclusion: Firstly, from the simulation results, we can found that the sales volume of online stores is increasing with the rising of the promotion level. But in the period of the “double eleven”, the sales promotion reached the maximum level, and the profit increased slowly, even had a decreasing trend. Secondly, we found that the consumers’ evaluations are decreasing with the increase of promotion effort. This may be due to the reducing of service quality caused by excessive promotion effort. The result of figure 8 shows that the number of consumers’ negative reviews has increased compared to the positive reviews. Hence the promotion effort may lead to negative e-WOM towards online store whereas the improvement in product quality can indeed bring positive e-WOM. Finally, from the cloud picture in figure 9, we can found that during the period of “double eleven,” consumers are more concerned about the service quality of the online store and the delivery speed. This finding further confirmed the previous findings that promotion effort may cause consumers’ dissatisfaction.
Abstract: In this paper, we build a causal interaction diagram between the factors that may influence the sales and profits of online stores. An online store’s real operation data were used to help determine the causal relationship between variables. Finally, we proposed a system dynamics model and conducted a simulation of the operation of online store. In this model, we focused on the impact of promotion and positive/negative e-WOM on the sales and profits of the online stores. The simulation results show a similar trend with the real data. And the main research finding shows that promotion is not a long-term measure for the sustainable development of online stores. Excessive promotion effort may lead to consumers’ dissatisfactions then bring to the increase of negative e-WOM. The systematic simulation can help us understand the long-term effect of promotion and e-WOM on the operation of online store better. Finally, we gave some management suggestions for the online stores’ sustainable operations.
Point 3: The heading for Section 6, ‘Patents’, seems strange. I would have expected the heading ‘Limitations and further research’.
Response 3: Thanks for your suggestion. We are sorry for the mistake in our manuscripts and we have changed the heading of Section 6 to “Limitations and further research”.
We hope that these revisions are satisfactory and that the revised version will be acceptable for publication in Sustainability.
Thank you very much for your work concerning my paper.
Wish you all the best!
Sincerely yours,
Zhou

Reviewer 2 Report
1. At line 111, in Table1, under the Row of Rate Variable.
Evaluation Decrease(ED) is defined as "The average evaluation decrease
of ..
2. At line 178, please describe more about how all the data obtained from the website. Do these technologies include text-mining or sentence crawling? And, the mechanism is how to identify positive or negative feedback.
3.At line 181, please write the version of SPSS.
4.At line 213, in equation(13), DPE(t) should be changed as PE(t).
5. In Figure 3, the authors compare simulation result with actual data meanwhile accuracy percentage, RMSEA or error rate had better be mentioned.(in line 308)
6. In line 248, please give one example to describe how the manager of line store constantly adjust the promotion strategies according to the situation in order to maximize profits.
7. Please add the measurement unit of vertical coordinate in Figure 6 and 8.
8. At line 289, Please describe the software or methodology which are applied to analyze the word frequency and get cloud pictures.
Author Response
Dear reviewer,
We are grateful to your professional suggestions. Based on these suggestions, we have made careful modifications on the manuscript. The modifications in our manuscript were all highlighted by using the revision mode of MS word.
Point 1: At line 111, in Table1, under the Row of Rate Variable. Evaluation Decrease(ED) is defined as "The average evaluation decrease of ..
Response 1: Thanks for your advice. We are sorry for the mistake in the original manuscript. And the word “increase” has been changed to “decrease”.
Point 2: At line 178, please describe more about how all the data obtained from the website. Do these technologies include text-mining or sentence crawling? And, the mechanism is how to identify positive or negative feedback.
Response 2: Thank you for your advice. According to your advice, we add the descriptions of data collection and sentiment analysis in the data collection section. The improved content is as follows:
We collected the reviews and sales data of a product from T mall supermarket daily. The data collection process was supported by crawler program with Python 3.5. The time span was from October 5th, 2017 to December 5th, 2017. The product is a kind of daily necessities with thousands daily turnovers. The system dynamic simulation is performed on Vensim. The DT was set to 1 day to simulate the sales and profits trend. The daily sales and the e-WOM maintenance are all the real data obtained from the website. Besides, the numbers of positive and negative e-WOM were based on the score calculated by Python sentiment analysis. If the score is greater than 0, the piece of e-WOM is regared as positive e-WOM, otherwise, negative e-WOM. In the simulation model, we input the data about daily promotion effort based on the real data and then observe the fluctuations of daily sales and accumulated profits. To determine the correlation coefficient between variables in the model, we firstly normalized the data, and then use SPSS to do a regression analysis. Finally, we compare the simulation results with the real data to verify the effectiveness of the model.
Point 3: At line 181, please write the version of SPSS.
Response 3: Thanks for your suggestion. We are sorry for the negligence in the original manuscript. The version of SPSS is 23 and we accordingly added the description in the manuscript.
Point 4: At line 213, in equation(13), DPE(t) should be changed as PE(t)
Response 4: Thank you very much for your suggestion. We have revised the false equation in the manuscript.
Point 5: In Figure 3, the authors compare simulation result with actual data meanwhile accuracy percentage, RMSEA or error rate had better be mentioned. (in line 308)
Response 5: Thanks for your suggestion. We have added the relevant index in the revision manuscript.
Point 6: In line 248, please give one example to describe how the manager of line store constantly adjust the promotion strategies according to the situation in order to maximize profits.
Response 6: Thanks for your suggestion. We add an example for the management of online store in Section 4.3.2 and give more details in Section 6. For your convenience, we list it below.
Section 4.3.2: For example, if the profits of the online store start to decrease during the promotion period, the promotion should be stopped in time and the improvement should be made refers to the e-WOM of consumers.
Section 6: From the long-term perspective, promotion is not suitable for the sustainable development of online stores. For online stores, they should keep eyes on the consumer e-WOM constantly, especially negative e-WOM. According to the feedback of consumers, online stores can discover their shortcomings and make corresponding improvement. Since the sales and profits are directly relevant with the consumer e-WOM, the product quality and service quality should be focused on.
Point 7: Please add the measurement unit of vertical coordinate in Figure 6 and 8.
Response 7: Thanks for your suggestion. We have added the unit of vertical coordinate in all the figures of our paper.
Point 8: At line 289, Please describe the software or methodology which are applied to analyze the word frequency and get cloud pictures.
Response 8: Thanks for your suggestion. We have added the descriptions of the methodology to analyze the word frequency and get cloud pictures in Section 4.3.3. For your convenience, we list it below.
We firstly used the third-party package (jieba) in Python to segment consumers' reviews, and then count word frequency. Besides, the word cloud picture is drawn by the wordcloud package in Python, see in Figure 9.
We hope that these revisions are satisfactory and that the revised version will be acceptable for publication in Sustainability.
Thank you very much for your work concerning my paper.
Wish you all the best!
Sincerely yours,
Zhou
